# Chunky Graphite in Low and High Silicon Spheroidal Graphite Cast Irons–Occurrence, Control and Effect on Mechanical Properties

**DOI:** 10.3390/ma13235402

**Published:** 2020-11-27

**Authors:** Jon Sertucha, Garikoitz Artola, Urko de La Torre, Jacques Lacaze

**Affiliations:** 1Azterlan, Basque Research Technological Alliance, Aliendalde Auzunea 6, 48200 Durango, Spain; gartola@azterlan.es (G.A.); udelatorre@azterlan.es (U.d.L.T.); 2Centre Inter-universitaire de recherche et d’Ingénierie des matériaux, CIRIMAT, Université de Toulouse, 4 Allée Monso, BP 44362, 31030 Toulouse, France

**Keywords:** spheroidal graphite, chunky graphite, spiky graphite, cast iron, tin, antimony, mechanical properties

## Abstract

Chunky graphite appears easily in heavy-section spheroidal graphite cast irons and is known to affect their mechanical properties. A dedicated experiment has been developed to study the effect of the most important chemical variables reported to change the amount of chunky graphite, namely the content in silicon and in rare earths. Quite unexpectedly, controlled rare earths contents appear beneficial for decreasing chunky graphite when using standard charge materials. Tin is shown to decrease chunky graphite appearance and it is evidenced that this effect is not related to rare earths. Finally, the effect of tin and antimony are compared and it is noticed that both suppress chunky graphite but also lead to some spiky graphite when no rare earth is added. Chunky graphite negatively affects the room temperature mechanical properties, though much more in the case of low silicon spheroidal graphite cast irons than in high silicon ones. Spiky graphite has been found to be much more detrimental and should thus be avoided.

## 1. Introduction

Spheroidal graphite cast iron has rapidly become one of the most important engineering materials after its discovery some 70 years ago [1]. Since then, the casting of spheroidal graphite cast irons has been successfully carried out by adding to the base melt an appropriate quantity of magnesium whose first action is to decrease the amount of sulfur and oxygen. Calcium and rare earths (RE) may be added together with magnesium for supporting this purification treatment with the further advantage of lowering the necessary amount of magnesium and thus the pyrotechnic effect of the spheroidization treatment. Spheroidal graphite is effectively obtained when the final amount of dissolved magnesium is about 0.020–0.025 wt.% [2,3] while compacted graphite corresponds to a level of 0.008–0.017 wt.% [4] in casting sections lower than about 60–70 mm.

Even if the spheroidization treatment has been carried out correctly, the presence of trace elements in the charge materials for the preparation of the melt may hinder the production of spheroidal graphite and then might affect the service properties of the cast parts. Such impurities lead to spheroidal graphite degeneracy known as spiky, crab-like and eventually mesh graphite as previously reviewed [5]. These different kinds of degenerate graphite first appear as elongated protrusions growing out from the spheroids, suggesting that they develop in the prismatic ***a*** direction of the graphite as does lamellar graphite. In his pioneering work on the degeneration of spheroidal graphite, Thielemann [6] quantified the deleterious effect of Al, As, Bi, Pb, Sb, Sn and Ti, but many other trace elements have been identified, see the review by Reynaud [7]. It has been realized since a long time that adding rare earths (RE) offsets the problem in most cases [8,9,10] which is thought to be firstly due to RE catching up impurities as stable compounds [11,12,13]. Amongst others, Zhao Bofan and Langer [11] did effectively evidence Ce-Sb compounds using melts that contained high levels of these elements (>500 ppm), see the review by Javaid and Loper [13].

The effect of these trace elements becomes more and more evident as the casting size is increased which seems to have been first related to the fading of the spheroidizing and inoculation treatments because of the lengthening of the solidification time [13]. An obvious action would thus be to increase the amount of magnesium and RE but this leads to the so-called over-treatment which has eventually been associated with the occurrence of a specific form of degenerate spheroidal graphite, namely chunky graphite [13,14]. From a metallurgical point of view, the main feature of chunky graphite is that its overall growth direction is along the basal ***c*** direction of graphite as it is for spheroidal graphite. Thus, the growth mechanism of chunky graphite certainly differs from the one mentioned above for protrusions. From a practical point of view, chunky graphite appears either as interconnected thin strings of graphite in large cells or as coarser strings in the last to freeze areas [15]. On 2D metallographic sections, chunky graphite may easily be confused with other graphite degeneracies and, in particular, compacted graphite [10] as shown with the wording “grouped vermicular” graphite used by Hoover [14]. What is anyway clear is that it is a cause of rejection of castings because it is expected to lead to a significant decrease of the mechanical properties, see the recent review by Baer [16].

Amongst the various factors that increase the risk of chunky graphite are casting size (or low cooling rate), high silicon content and over-treatment with spheroidizers, in particular, RE as reviewed a long time ago by Hoover [14] and again demonstrated since then by Asenjo et al. [17] although rejected by Gagné and Argo [18]. An increase in the casting size does enhance all types of spheroidal graphite degeneracies, and this may be due in part to the fact that nodules are fewer and larger. This led to the suggestion of enhancing graphite nucleation by higher cooling rates or increased inoculation. While the effect of increased cooling rate, in particular by the use of chills, seems clear and accepted [16], the beneficial effect of increased inoculation is controversial [19,20,21] though it is still considered [22,23]. That fewer and larger nodules are more prone to degenerate is well known, this has been recently evidenced by X-ray computerized tomography (4D-XCT) [24] and theorized for the case of exploded graphite [25]. Finally, using the fact that chunky graphite precipitates during coupled growth with austenite, it has been suggested that the effect of increased silicon could be related to a change in the austenite/liquid surface energy [26].

The practical way to counteract the detrimental effect of RE is to add antimony, though other elements have been mentioned such as As, Pb and Sn [19]. It is of prime interest to stress that these trace elements are otherwise considered as promoting degeneracy of spheroidal graphite when present above a critical level and should be balanced by RE addition in that case. This ambivalent role of trace elements and RE is quite striking and would lead to the conclusion that chunky graphite should not appear if pure charge materials and no RE were used for melt preparation. That this is so has been recently reported by Jia-xin Bai [27] for low-silicon cast irons prepared with high-purity charge materials, but such a way certainly leads to a dramatic cost increase. It remains unclear if an optimum addition of elements such as Sb and Sn could be found that would be below the critical limit for graphite degeneracy when no RE are present, but high enough for eliminating the risk of chunky graphite.

Although the effect of silicon on the formation of chunky graphite has long been documented [15,19,28,29,30], much less is known about its appearance in the high-silicon spheroidal graphite cast irons that are currently under development [26,31]. Such an interest is also triggered by the fact that it has been demonstrated that chunky graphite does not affect room temperature mechanical properties of high-silicon spheroidal graphite cast irons contrary to the case of low-silicon alloys [32]. Accordingly, the present investigation was dedicated to: (i) the development of an instrumented small scale casting giving low cooling rate to generate conditions prone for chunky graphite formation; (ii) further investigating the effect of high Si as compared to standard Si content, and the effect of a low level of RE and Sb on chunky graphite occurrence; (iii) documenting the effect of Sn in eliminating chunky graphite and confronting it to that of Sb; (iv) characterizing the impact of chunky graphite on mechanical properties of low and high silicon spheroidal graphite iron.

## 2. Experimental

For studying the formation of chunky graphite in a foundry laboratory, the basic idea was to define a casting with a size as small as possible but with a long enough solidification time to promote heavy section castings behavior. A first trial (P1) was performed using an exothermic mold (based on an exothermic feeder) but an important part of the melt together with slag came out of the mold just after pouring. Due to this fact, exothermic molds were rejected and were replaced by isolating ones with satisfactory results. Nine further experimental trials have then been carried out which are chronologically denoted by the trial number following the capital letter “P”. Table 1 shows a summary of the trials.

The melts were prepared as described previously [26] in a medium frequency induction furnace. When needed a FeSi75 alloy was used for adjusting the silicon content. The base melt held in the furnace was transferred at about 1500 °C to a ladle with a chamber in which FeSiMg spheroidizer and inoculant were located and covered with steel cuttings. The amount of spheroidizer and inoculant was 1.2–1.3% and 0.30% of the melt weight, respectively. Two FeSiMg alloys and two inoculants have been used that contain or not some RE. The inoculant without RE was the same FeSi75 alloy as the one mentioned above, after crushing and proper size selection (0.2–0.7 mm). A commercial product, here after denoted as Anoc, was used as RE-bearing inoculant. The compositions of all these treatment alloys are listed in Table 2. Intermediate RE levels were obtained by appropriate mixing of products.

Once the Mg treatment had finished, a chemical composition sample was obtained and analyzed by spark emission spectroscopy for controlling and validating that the target composition was obtained. The alloy was then poured in two thermal analysis (TA) cups (for reproducibility) and in the mold illustrated in Figure 1. This mold was instrumented with four Ø3 mm stainless steel sheathed K-type thermocouples whose locations are indicated in the figure. These thermocouples were covered with a further external ceramic sheath which was in contact with the alloy cast into the molds. Exceptionally, K-type thermocouples with a metallic sheath 1.5 mm in diameter were employed in P8 trial while keeping the same experimental method. Cooling curves from the TA cups and from the thermocouples were recorded in all cases.

After cooling down, the TA cups and the cylindrical castings were cut and prepared for metallographic analysis to check their microstructure and to determine the area fraction of the different graphite particles present in the central zone of the TA cups (three different fields per cup) and in locations close to the tip of the four thermocouples in the cylindrical castings (five different fields in each location). The magnification of all images used in these analyses was ×100 corresponding to an area of 0.757 mm^2^ each.

The micrographs were then binarized and all particles with an area less than 25 µm^2^ were disregarded. The total count, N_count_, and area fraction, f_A_, of graphite particles were then evaluated. Then, the different types of graphite particles were sorted into three categories following criteria described in Table 3. The circularity and aspect ratio are respectively defined as 4π·$\frac{A}{P^{2}}$ and $\frac{L_{\max}}{L_{\min}}$, where A is the area of a given graphite particle, P is its perimeter, and L_min_ and L_max_ are its minimum and maximum lengths, respectively.

Types V and VI relate to regular and highly regular nodules, respectively, while type III corresponds to chunky graphite in most cases (trials P2 to P5 and P7), except in the case of addition of Sn or Sb when it was also used to quantify spiky graphite (trials P6, P8 and P9). In trial P10, both chunky graphite and spiky graphite were present, so their area fractions were separately determined. This differentiation between chunky and spiky graphite was performed by means of visual inspection after the type III particles had been extracted by automated image analysis. The area fractions of graphite type III, V and VI were evaluated and then expressed as the relative value of the total graphite area fraction, f_III_A_, f_V_A_ and f_VI_A_, with f_III_A_ + f_V_A_ + f_VI_A_ = 1. After graphite characterization, the samples were etched with Nital for evaluating the constitution of the matrix, i.e., the fraction of pearlite and ferrite.

A representative piece of each produced cylindrical casting was also used to determine the final composition of the alloys. Carbon and sulfur contents were determined by combustion (LECO CS300) while the inductively coupled plasma mass spectrometer, ICP-MS, (Agilent 7500ce) was used for other elements after dissolving a metallic sample in a mixture of acids. Finally, silicon contents were determined by gravimetric techniques.

For mechanical testing, a series of eight additional castings were prepared with the mold shown in Figure 1 but without thermocouples. These castings will be referred to as PM1-PM8 in Section 3.4. All tensile tests were performed on specimens with a gage 5 mm in diameter and 30 mm in length with threaded ends (Figure 2). On these specimens, the ultimate tensile strength (UTS), the elongation (A) and the yield strength (YS) values were measured using a Zwick Z100 tensile testing equipment at a constant strain rate of 2.5 × 10^−4^ s^−1^.

## 3. Results and Discussion

Table 4 contains the results of the chemical analyses carried out on samples obtained from the produced cylindrical castings. It was verified that the medal sample and the casting itself gave very similar chemical analysis results, thus ensuring that the whole casting procedure was correct. Values of the carbon equivalent (CE) that were determined by using the expressions proposed by Castro et al. [33], CE_99_, and in ASM handbook [34], CE_ASM_, have been also included in Table 4. The calcium and bismuth contents were checked also and found below 0.0010 wt.% and 0.0005 wt.%, respectively, in all cases.

### 3.1. Effect of Si and RE Contents

The first four castings (P2 to P5) were prepared for studying the effect of silicon (2.3 and 4.0 wt.%) and that of RE when added at the level of 100–150 ppm. Figure 3 illustrates the temperature recorded with the thermocouples T1–T4 in casting P5. These records were similar for all four trials P2–P5 and show that solidification proceeded in two steps, namely a pro-eutectic reaction and then a eutectic plateau. The pro-eutectic reaction is hardly visible for T4, while it is clearly sensed with all three other thermocouples. While for T1 it leads only to a slope change, the pro-eutectic thermal arrest shows recalescence for T2 and T3 suggesting equiaxed solidification of austenite in these two locations. It may thus be postulated that T4 is in an area with a significant temperature gradient where austenite grows as columnar dendrites, T1 at a location where the temperature gradient is lower but still present, while T2 and T3 are within or close to the thermal center of the casting. In relation with this, it is seen that the eutectic recalescence is the highest for T2 and T3, lower in T1 and is very limited in T4. The above description agrees with usual observation and analysis of casting solidification, see e.g., Dioszegi et al. [35].

The results obtained from the metallographic characterization are reported in Table 5. The total area fraction, f_A_ (%), and total particle count, N_count_ (mm^−2^), of graphite are listed as well as the relative area fraction of each type of graphite. For these four castings, all graphite of class III was associated with chunky graphite. Chunky graphite was observed in the four locations T1–T4 of all cylindrical castings P2–P5, though in an amount significantly higher for P2 and P3 than for P4 and P5 in relation with the higher silicon content in these former castings. When analyzing the microstructure, it was noted that the amount of chunky graphite could in several cases vary a lot within the 3 (TA cups) or 5 (cylinders) measurement fields made on a given location. This led to the standard deviation being sometimes equal or even higher than the mean value. This suggested indicating between brackets the minimum and maximum values of f_III_A_. Such variability affects also the number of class V and class VI spheroids and also the total count of particles, N_count_. The standard deviation for this latter is given for illustration.

It must be remarked that the quantification of chunky graphite can be troublesome due to the fact that chunky cells have millimeter size, i.e., a size similar to the size of the micrographs used in the present work. Using lower magnification micrographs would have led to other difficulties such as not separating properly the various shapes of graphite. In practice area fraction of class III graphite will be measured with a low standard deviation if it is either higher than 0.80 or lower than 0.20, but the scattering could be very high for intermediate values. Once this was realized during the first characterizations for this study, it was decided to select the location of the three (resp. five) micrographs: one should show the highest amount, one the lowest one and the third (resp. last three) should be representative of the area under investigation. In this way, it was thought that the error on the average value would be minimized.

Owing to the much higher cooling rate, the thermal cups showed no chunky graphite for P4 and P5 and a low amount for P2 and P3. Thus, as expected, they are of little use for assessing the tendency of a melt to give chunky graphite. Nonetheless, it was of interest to notice a change in distribution. The micrographs in Figure 4 illustrate the graphite distribution in casting P2 for the location T3 and for the central part of the TA cup. It is seen that the decrease in the chunky graphite fraction corresponds to a significant change in its distribution, with large cells of interconnected strings replaced by isolated coarser strings. Cells are expected to have appeared early during the eutectic reaction while isolated strings relate to areas where solidification ended [15]. Finally, in low silicon alloys, some rare special features were observed that are illustrated in Figure 4c. The shape suggests a two-phase growth around a dendrite arm.

The area fraction of chunky graphite particles classified as type III has been plotted in Figure 5 for the four thermocouple locations and the TA cups of all P2–P5 castings. The most relevant chemical information is also indicated for each casting to ease reading. It is seen that:Increasing the silicon content from 2.3 wt.% to about 4.0 wt.% leads effectively to a significant increase of f_III_A_.Though TA cups show little or no chunky graphite, they appear sorted as the castings are. The melts giving the highest f_III_A_ values in cups are also those giving the highest f_III_A_ in the cast cylinders.For both high and low silicon alloys, the addition of RE to alloys free of these elements shows a decrease in chunky graphite. Furthermore, this effect seems stronger for the low silicon alloys than for the high silicon alloys. This beneficial effect of RE goes against the trend generally considered but must be associated with the low level of the addition and has been observed previously [26]. A possible reason is that RE counteract impurities that are at very low levels such as Al, P, S and Ti, see Table 4.

In Figure 3, it appeared that the cooling curves recorded with thermocouples T2 and T3 are capable of sensing effects that were not recorded with thermocouples T1 and T4. Owing to the larger changes in chunky graphite fraction seen in location T3 (Figure 5), this thermocouple was selected for comparing the cooling records for all four castings and this is shown in Figure 6. The corresponding chunky graphite fractions have been indicated in the figure caption.

In cast iron, the equilibrium eutectic temperature depends firstly on the alloy’s silicon content, increasing by about 4.246 °C per wt.% of silicon [33,36]. Hence, high-Si castings P2 and P3 should have a very similar equilibrium eutectic temperature whose value is higher by about 7 °C than that for low-Si castings P4 and P5. It is seen that the sorting of the curves in Figure 6 agrees with this predicted change in eutectic temperature. Further, it is noticed that the maximum plateau temperature, T_ER_, increases with the chunky graphite fraction. This increase is in agreement with the fact that the growth rate of chunky graphite cells is much higher than that of spheroidal graphite eutectic cells, being in between undercooled and lamellar graphite growth rates according to Källbom et al. [37].

### 3.2. Effect of Intermediate Level of RE and of Antimony

It has been seen in the previous section that adding some RE decreased the amount of chunky graphite though its level remained quite high. This prompted a trial with an intermediate level of RE in an attempt to find an optimum value for high silicon alloys. In effect, Skaland [38] has reviewed a number of works suggesting there is an optimum RE content for either graphite shape and nodularity or nodule count, and it was wondered if such a relation could be similarly found for chunky graphite. The amount of RE that was targeted was half that added in P2 and the corresponding trial has been numbered P7. The total amount of RE in P7 is about 50 ppm close to the value of 60–80 ppm achieved in a previous study [26] and slightly lower than half the 110–165 ppm in P2 and P4.

The results of the metallographic characterization of casting P7 are given in Table 6. As before, minimum and maximum values for f_III_A_ and standard deviation for N_count_ are listed which show marked variability in some locations. By comparison with trials P2 and P3, it was observed that the intermediate value of RE did effectively decrease the amount of chunky graphite in the locations T1–T3 but surprisingly not in T4, see Figure 7. It is also noticed that the amount of chunky graphite remains quite high even in the cups where 10% of the graphite is classified as chunky. In this respect, it may be noted that the total amount of deleterious elements, Al, P, S and Ti, at 330 ppm is far above the amount of RE measured in casting P7. Accordingly, it seems unlikely that there exists an optimum amount of RE that would allow suppressing chunky graphite.

The above conclusion as well as the literature review suggested another trial consisting in adding antimony at the usual level used to avoid chunky graphite but without adding RE. This goes in line with the observation that a very low level of Sb improves nodularity in RE-free standard spheroidal graphite cast irons [2,39,40,41] though there is a risk of appearance of spiky graphite if the Sb addition is higher than 100 ppm [13] or 200 ppm [42]. In the present work, an addition of 39 ppm of antimony without RE was done in casting P9, and this casting was found to be totally free of chunky graphite. However, some spiky graphite appeared instead which is illustrated with the micrographs in Figure 8. The metallographic values are listed in Table 6 and plotted in Figure 7 where it is seen that up to 20% of graphite is spiky in location T3.

That antimony suppresses chunky graphite even if no RE is present demonstrates that its action is not only to fix excess RE when present as most often assumed. Further, that this finding applies to high silicon cast irons which are so prone to chunky graphite formation suggests that even a low level of antimony, as the 39 ppm added here, is sufficient to strongly affect the growth mechanism of spheroidal graphite. As above, one may wonder if an optimum addition of antimony could be found which would lead to no chunky graphite and no spiky graphite. However, any attempt to find an optimal addition of antimony at some tens of ppm would not be of practical use and this was thus not carried out.

### 3.3. Effect of Sn on Chunky Graphite in High-Silicon Cast Irons

In contrast with antimony, tin addition for suppressing chunky graphite in low silicon cast irons has been reported to be much higher at some 250–500 ppm in 5–8 inches side cubes [19]. Considering the detrimental effect of silicon, three RE-free high silicon castings were prepared to study the effect of Sn at the level of 250 (P10), 500 (P8) and 1000 (P6) ppm. Table 7 gives the full record of data of graphite characterization in these castings which shows that chunky graphite totally disappeared at 0.051 wt.% (casting P8) and 0.094 wt.% (casting P6) of Sn but also that spiky graphite appeared in all castings with added Sn. In case of P10 at 230 ppm Sn, both chunky graphite and spiky graphite were observed. Note again that the scattering is quite high for some of the measurements. These results are compared in Figure 9 to values for casting P3 without RE and no Sn.

Looking at Figure 9 suggests comparing the microstructures in locations T1–T3 where both chunky graphite and spiky graphite formed in casting P10 to location T4 where only spiky graphite was observed in the three castings with added Sn. This comparison is illustrated in Figure 10 with micrographs in location T1 in the column to the left and in location T4 in the column to the right. Though this may need further investigation, it seems that spiky graphite appeared late during solidification in all three castings and both locations. In location T1 of casting P10, chunky graphite seems to have precipitated at an intermediate solidification stage. Such characteristics are guessed to relate to microsegregation building up during solidification.

Whatever the amount of added Sn, it is of interest to notice that the highest amount of spiky graphite that has been measured is much lower than when as little as 39 ppm of antimony was added in casting P9. Furthermore, based on Figure 9 and the micrographs in Figure 10, the intermediate level of Sn at 500 ppm would appear satisfactory for avoiding chunky graphite and minimizing spiky graphite.

Pursuing the above cooling curve analysis, Figure 11 compares the cooling curves at T3 for castings P3 without the addition of Sn and P6 at 0.094 wt.% Sn. The decrease in the temperature of the first arrest associated with precipitation of austenite is in line with the change in the carbon equivalent of these alloys, but the more striking feature is that the eutectic plateau differs significantly. For P3 which has been shown to present 87% of chunky graphite, there is a marked recalescence as already pointed out. With the addition of 0.094 wt.% Sn in P6, this recalescence has nearly disappeared while the temperature of the start of the eutectic plateau is the same as in P3. Due to the use of different thermocouples for P8, the cooling curve at T3 cannot be compared because the eutectic was recorded at a lower temperature, but again it did present only very little recalescence in relation to the low level of chunky graphite found in this casting.

The whole set of available data of chunky graphite fraction versus recalescence for the cylindrical castings is plotted in Figure 12. As expected, and described above, there is a clear trend for an increased recalescence when chunky graphite becomes more and more prevalent. However, there are outliers on or close to the axes which would hinder using such a relationship for prediction purposes, apart maybe if considering the thermocouple located in T1. It may be worth noting that the maximum recalescence recorded at the center of square blocks with a 5 cm thermal modulus and solidifying in 2.5 h was twice as much at 8 °C [43] than the value recorded in the cylindrical castings of this study. This is a clear indication that even with the molds used in the present study local thermal effects are very much dependent on the heat transfer conditions of the whole casting.

This result was a little bit disappointing and led us to look more precisely at those points in Figure 12 with no recalescence and showing a high fraction of chunky graphite. The two highest values are associated with casting P7 and moreover the highest of them relate to location T4 which is expected to be in a well-established temperature gradient and thus should be associated with a decreased amount of chunky graphite according to the literature review. As a matter of fact, Figure 7 shows that another casting, namely P3, presented also a very high level of chunky graphite in position T4. The first idea was that the heat transfer conditions were different for these two castings than for all others. To verify this, the cooling curves were compared and are plotted in Figure 13. Unexpectedly, these cooling curves differ significantly from each other, apparently because the pouring temperature differed significantly. Thus, the cooling curves do not give any hint that could explain why these two castings P3 and P7 showed so much chunky graphite in location T4. On the other hand, this comparison points out an important fact: even in a positive temperature gradient—the cooling curve for P7 does certainly correspond to such conditions—chunky graphite can develop during the whole eutectic solidification process. In other words, the use of chills may not always be a remedy against the formation of chunky graphite.

To explain the maximum in chunky graphite fraction at location T4 in castings P3 and P7, the only correlation that we found is that these are the two alloys with the lowest CE values of the series, 4.11 and 4.05 wt.% respectively. Accordingly, the eutectic reaction takes place while an austenite dendrite network is well established, and this may be particularly true within a temperature gradient. This certainly points out the conditions for the early growth of graphite which finds some support in the schematic proposed by Zhou et al. [44] who suggested that chunky graphite nucleates at the interface between austenite and liquid where the carbon content is highest.

One important output of the present study is that Sn acts as Sb in suppressing chunky graphite formation even when RE are not involved. There is however a major difference between these elements in the needed level to be effective. While 20–50 ppm of Sb appear sufficient, ten times more are needed if Sn is employed. From a practical point of view, it may be expected an easier control with tin than with antimony. From a theoretical point of view, this strongly suggests that the mechanism by which these elements affect graphite growth differs from each other. The risk when using either Sn or Sb is to favor pearlite formation, see Appendix A.

At the highest amount of Sn they added, namely 1000 ppm, Karsay and Campomanes [19] noticed the presence of intercellular inclusions and assumed this could affect mechanical properties. Similarly, it has been shown that spiky graphite may be even worse than chunky graphite [45]. This triggered the mechanical tests that are presented in the next section.

### 3.4. Mechanical Properties

The above microstructure results suggested studying the effect of chunky graphite and of spiky graphite on the mechanical properties. To ascertain the findings, it was thought useful to compare results for both low-Si and high-Si spheroidal graphite irons. Accordingly, four alloys were planned for each type of cast iron, with the intention to have: (1) fully spheroidal graphite; (2) a medium quantity of chunky graphite; (3) a high quantity of chunky graphite; (4) some spiky graphite. Eight new castings were thus produced with the same mold as before (Figure 1) but with no thermocouples, denoted as PM1––PM8. Five samples for the tensile test were machined out from each casting at locations 1–5 as shown in Figure 14. Table 8 shows the achieved values for composition and microstructure characteristics, while the whole compositions are listed in Appendix B. The metallographic characteristics have been determined in a cylindrical cross-section positioned as close as possible to the fracture surface of the tensile specimen.

In Table 8, the results for the two specimens from casting PM1 with no CHG or spiky graphite but with elongation below 6% were greyed due to the presence of defects in the fracture surface. Hence, these samples were removed from further analysis. Looking at all other results in Table 8, it is seen that for each casting the UTS and YS values vary little. However, it is also evident that there is an important scattering of the A values.

For analyzing this scattering, it appeared of interest to look first at the best of the tensile curves for each of the castings, this is done in Figure 15 for low-silicon (a) and high-silicon (b) alloys. Along each of these curves, the amount of chunky graphite or spiky graphite is indicated. For both types of alloys, it is seen that spiky graphite fractions above 0.11–0.12 are highly effective in decreasing the elongation at rupture. Chunky graphite is far less effective on high-silicon alloys, while the results for low-silicon alloys need further analysis, see later. It must be noted that PM5-1 showed a 100% pearlitic microstructure in the spiky graphite areas and 40–50% of pearlite in the rest of the inspected cross sections, thus explaining its higher tensile strength. The presence of pearlite also contributes to the decrease in the elongation at rupture together with the spiky graphite.

Focusing on high-silicon alloys, Figure 16 shows the change in UTS (a) and A (b) as a function of f_III_A_ for the alloys investigated in the present work. In this figure, our results are compared with those of two studies that investigated the effect of chunky graphite in an extended domain of degeneracy. Nakayama et al. [46] have studied this effect with fully ferritic cast iron at 3.5 wt.% Si that was cast as a large cylinder. The samples were machined out from this cylinder in locations having chunky graphite fraction varying between 0% and 82%. Källbom et al. [47] prepared five melts denoted A to E with 3.2–3.4 wt.% Si that were cast in a mold with plates of varying thickness. The chunky fraction changed from 0% in the thin plates to 100% in some of the thicker plates. The UTS and A values from these two studies were picked up on the provided graphs, excluding those results where shrinkage was evidenced by the authors. These values are reported in Figure 16 as a function of the chunky graphite fraction which was measured in both cases on the fracture surfaces and thus probably over-estimated.

In the graphs of Figure 16, the values of a cast iron with 4.2 wt.% Si and no chunky graphite from previous work [32] are also reported, namely 580 MPa for UTS and 15.4% for A. It is thus seen that 30% of chunky graphite decreases the UTS value to about 540 MPa, but that a further increase in chunky graphite has no effect. This agrees with Nakayama et al. [46] who suggested an abrupt decrease in UTS between 20% and 40% chunky graphite, while the results by Källbom et al. [47] suggest a slight but continuous decrease. In contrast, it is most often reported that chunky graphite does not affect the yield stress, though the results by Nakayama et al. [46] and Källbom et al. [47] show a very slight decrease. The present YS values for the high-silicon alloys vary between 425 MPa and 480 MPa with a mean at 460 MPa which is only slightly lower than the previously reported value of 480 MPa [32].

Figure 16b shows the change of A with the amount of chunky graphite. The value previously evaluated [32] for a cast iron with 4.2 wt.% Si and no chunky graphite was 15.4% and is indicated with the large red dot along the Y-axis in Figure 16b. All studies agree that 30–40% of chunky graphite lead to an abrupt decrease of A with respect to the value for a sound casting. Such a conclusion could not be drawn in our previous work but is now evidenced.

Finally, both graphs in Figure 16 show a strong effect of spiky graphite. Though the number of samples investigated during the present study is not very large, an attempt was made to sort the results in such a way that statistical trends could be deduced concerning the effect of chunky and spiky graphite, comparing low and high silicon alloys. The analysis was performed by defining sets of specimens depending on their content of chunky graphite and the presence of spiky graphite, which were compared by means of hypothesis testing on the variance and the average of a normal population. The population of the results matching each case of interest submitted to the hypothesis test was split into two sets with a variable number of samples each. The size of each sample depended on a threshold value for which a significant difference was found. The significance level was set to 5% in all cases.

To begin with the high silicon specimens with less than 8% of spiky graphite, all the sets that were tested gave a homogeneous outcome in terms of YS and UTS: the presence of chunky graphite at an amount higher than about 30% does not lead to a statistically significant variation of these two mechanical parameters. On the other hand, the presence of chunky graphite in high silicon specimens from Table 8 did affect elongation as observed by Källbom and Nakayama. Being more specific on this effect, it was found that a statistically significant difference of average elongation values is found when comparing the population with f_CHG_A_ values below 49% and the population with f_CHG_A_ values over 69%. The corresponding average values for the interval of 95% confidence are 6.7% ± 1.0% and 5.0% ± 1.1%, respectively. Thus, the observations conclude that high silicon ductile irons do not reduce its YS nor its UTS by the only effect of chunky graphite and that a threshold value can be established below which the effect of chunky graphite in the elongation is not significant.

Analyzing the rest of the dataset with no spiky graphite, YS strength showed to be independent of the chunky graphite content. Nevertheless, opposed to what was observed with high silicon alloys, UTS of low silicon alloys was affected by chunky graphite. This is evidenced when the set of specimens with f_CHG_A_ < 17% is compared to the set with f_CHG_A_ > 22%. In this case, the average UTS of each set is significantly different, giving values for the interval of 95% confidence of 395 ± 11 MPa and 342 ± 76 MPa, respectively. Using the same two sets for hypothesis testing with the elongation gave the same results in terms of significant variations of average elongation: 12.0% ± 2.2% and 4.0% ± 4.0% respectively. The confidence interval of the set with f_CHG_A_ > 22% has been narrowed since, given the high variance of the data, the population was too small and the Student’s t multiplier led to values without physical meaning.

With the information above as a reference, it can be stated that, for the material employed in this study, the high silicon ductile irons show more tolerance to the presence of chunky graphite than the low silicon ones. On one hand, UTS do not show any reduction on high silicon alloys due to chunky graphite when above about 30%, while low silicon alloys do. On the other hand, the sensitivity threshold at which the elongation dropped was close to 50% chunky graphite for the high silicon alloys while it was around 20% for the low silicon ones. Thus, the mechanical properties of these last group of alloys are affected by a smaller amount of chunky graphite in the microstructure. Furthermore, the average elongation drop for the high silicon alloys was 1.7% while it was 8% for the low silicon ones when the chunky graphite sensitivity threshold was exceeded.

To complement the results on the effect of chunky graphite, the tensile data on specimens containing spiky graphite were also analyzed, leading to the fact that even small spiky graphite contents cause a strong change in mechanical properties. Again, a threshold was detected for the high silicon alloys, as the results for the set of specimens with f_spiky_A_ values below 10% are not significantly different from the set with f_CHG_A_ < 49% and free of spiky graphite. On the low silicon alloys though, the mechanical properties varied significantly with the slightest content of spiky graphite. In both cases, high and low silicon alloys, the elongation drop that is caused by the presence of spiky graphite is caused by twice as much content in chunky graphite. Thus, as a rule of thumb, the results in Table 8 indicate that spiky graphite is twice as bad as chunky graphite when the sensitivity threshold is exceeded. This confirms the conclusion of Foglio et al. [45].

The concomitant decrease of A and UTS is illustrated in Figure 17 by plotting one versus the other for the high silicon alloys presently investigated (4.2 wt.% Si) and the ones studied by Nakayama et al. [46] and Källbom et al. [47] (3.2–3.5 wt.% Si). The two sets of data are clearly differentiated owing to the increase of the maximum UTS value with silicon. The shape of the evolution suggests comparing it with stress–strain curve and this has been done by superimposing the tensile test curve for sample PM8-5. This fact suggests that the presence of spiky graphite or chunky graphite in the high silicon ductile iron samples studied here lead to two distinct failure mechanisms. On the one hand, the five datapoints below 500 MPa correspond to samples containing spiky graphite but no chunky graphite, for which the specimens have failed in the elastic region of the tensile curve. On the other hand, the datapoints over 500 MPa correspond to specimens containing chunky graphite, for which the specimens have failed after elastic yield and with no apparent necking. Provided the matrix is similarly 100% ferritic for both cases, it is reasonable to also consider the toughness to be similar. This points to an elastic fracture mechanics failure driven by spiky graphite acting as the stress intensity raiser feature in the microstructure. The effect of chunky graphite as a stress intensity raiser is thus much less severe, allowing elastoplastic fracture mechanics to intervene, as specimen collapse is found well beyond yield strength.

## 4. Conclusions

A dedicated experiment mimicking heavy-section solidification has been developed to study the effect of the most important chemical variables reported to change the amount of chunky graphite, namely the content in silicon and in rare earths. Quite unexpectedly, controlled rare earth contents appear beneficial for decreasing chunky graphite when using standard charge materials. As does antimony, tin is shown to decrease chunky graphite appearance when added at a level of 500 ppm. It is also evidenced that this effect of tin and antimony is not related to rare earths which calls for further study on the role of these elements on graphite growth. However, if no rare earths are present, the addition of tin or antimony might lead to spiky graphite. Work is going on to determine if a critical level of tin at about 250–500 ppm would give sound castings, i.e., without chunky and spiky graphite. Chunky graphite affects the room temperature mechanical properties, though much more in the case of low silicon spheroidal graphite cast irons than in the alloys with high silicon contents. Spiky graphite is found to be much more detrimental than chunky graphite and its avoidance is considered critical. However, it has been observed that the mechanical properties of high silicon cast irons are not much sensitive to the exact amount of chunky graphite when above 30%.

## Figures and Tables

**Figure 1 materials-13-05402-f001:**
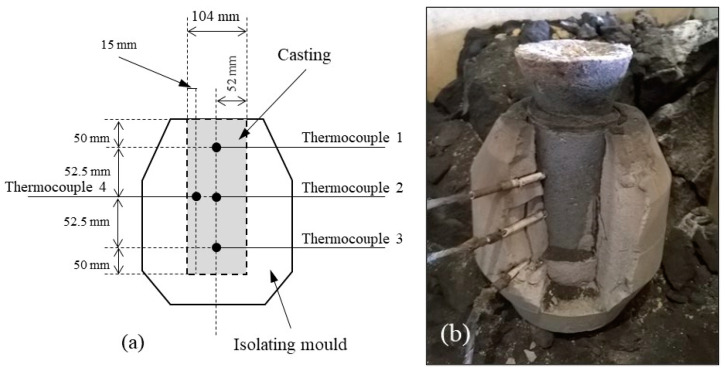
Schematic of the isolating mold with the indication of the location of the thermocouples (**a**) and a photograph of the mold just before removing the casting (**b**).

**Figure 2 materials-13-05402-f002:**
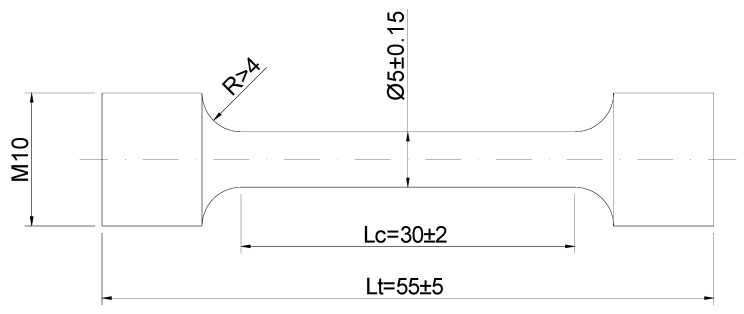
Sketch of the specimens employed in mechanical testing. Dimensions are given in millimeters.

**Figure 3 materials-13-05402-f003:**
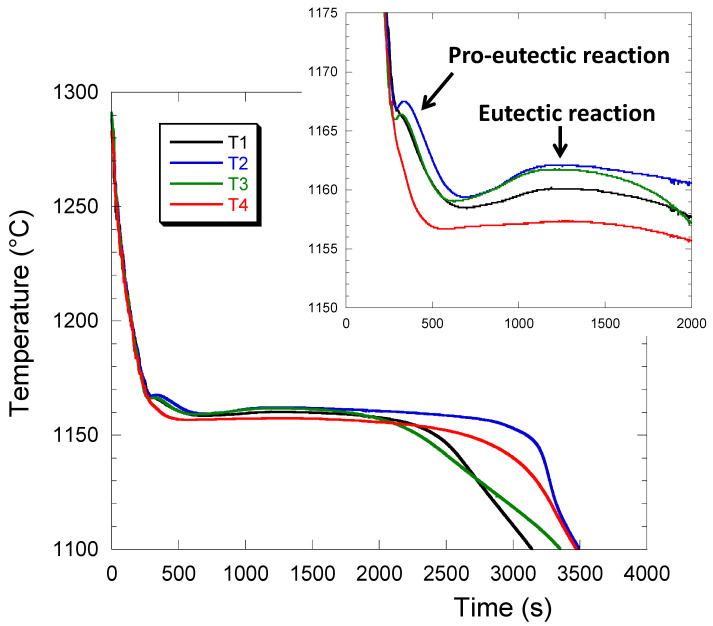
Example of the temperature evolution in the four locations T1–T4 of casting P5. The insert shows an enlargement of the beginning of solidification.

**Figure 4 materials-13-05402-f004:**
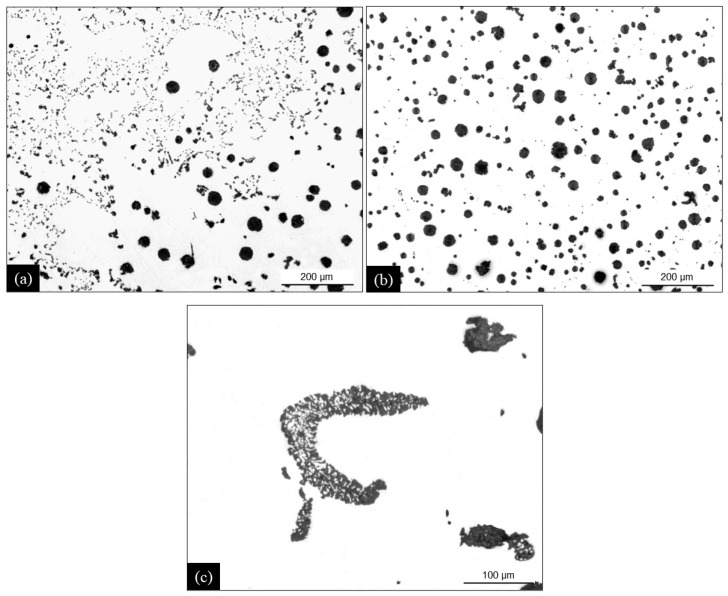
Micrographs of high-silicon casting P2 in location T3 of the cylindrical casting (**a**) and in the center of the thermal cup (**b**). Complex graphite string in location T3 of low silicon casting P4 (**c**).

**Figure 5 materials-13-05402-f005:**
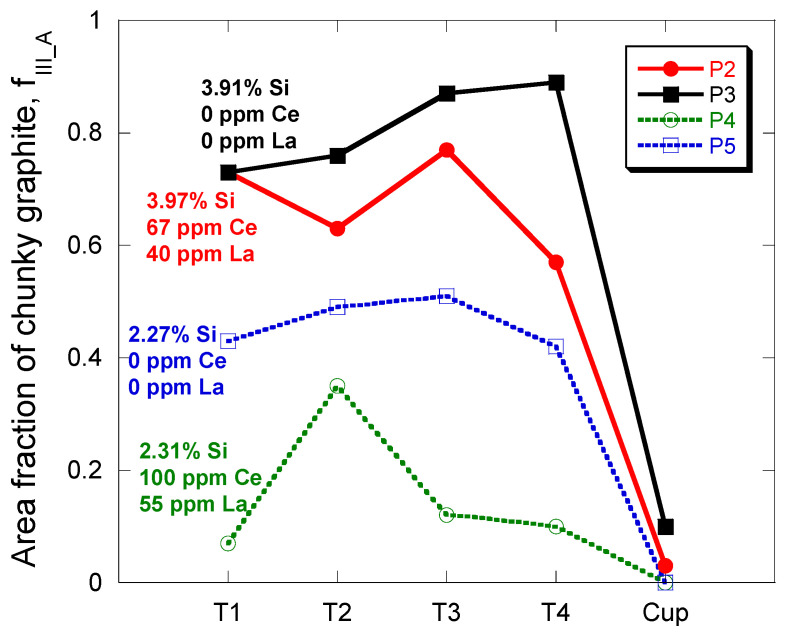
Evolution of the chunky graphite fraction f_III_A_ with the chemical composition for the four thermocouple locations in the cylindrical castings and for the thermal analysis (TA) cups.

**Figure 6 materials-13-05402-f006:**
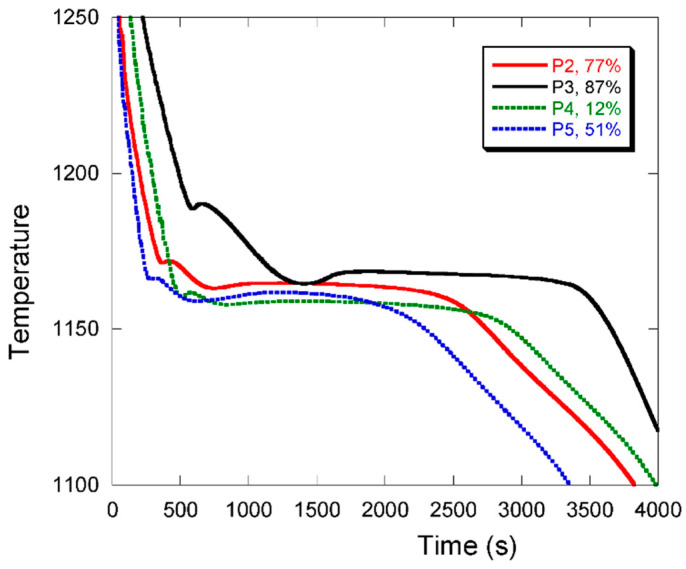
Cooling curves were recorded by thermocouple T3 in the four castings P2–P5. The corresponding fractions of chunky graphite, f_III_A_, are indicated in the caption.

**Figure 7 materials-13-05402-f007:**
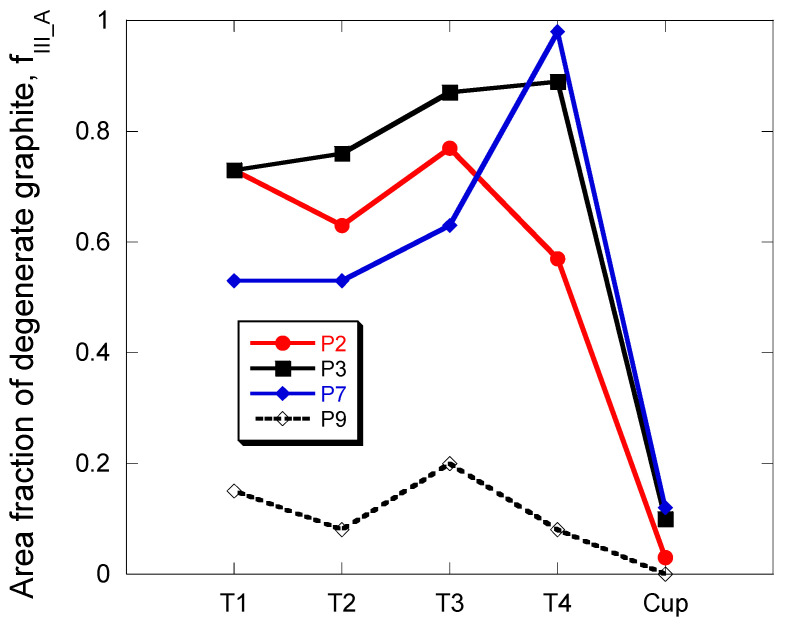
Evolution of f_III_A_ for the four thermocouple locations in the cylindrical castings P2, P3, P7 and P9 and for the TA cups. In P9, this is spiky graphite that was observed instead of chunky graphite.

**Figure 8 materials-13-05402-f008:**
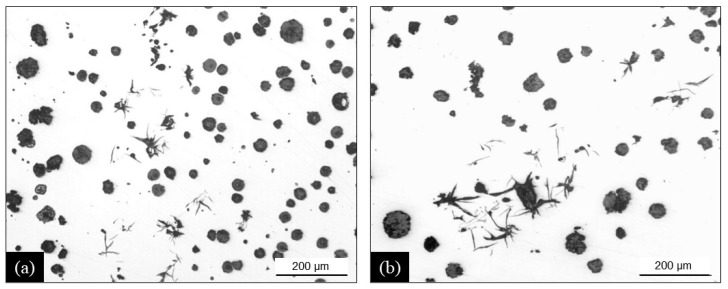
Micrographs illustrating the appearance of spiky graphite due to the addition of Sb in casting P9, locations T1 (**a**) and T3 (**b**).

**Figure 9 materials-13-05402-f009:**
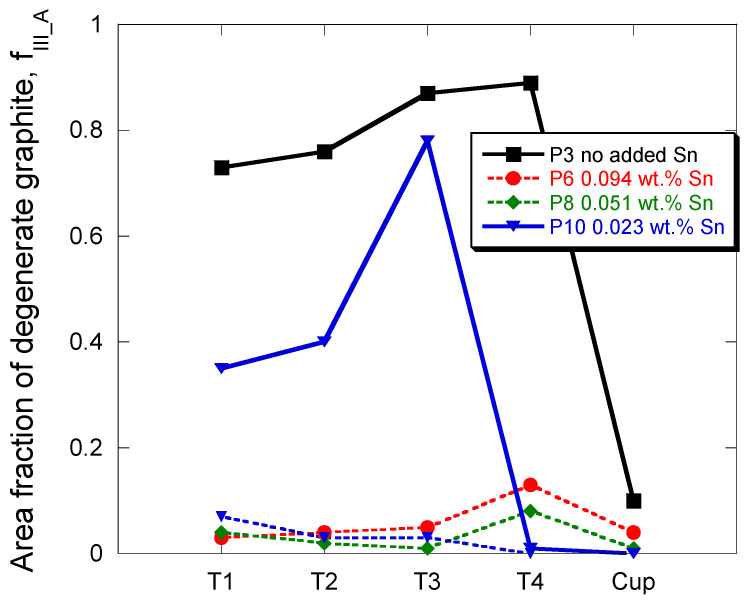
Evolution of f_III_A_ for the four thermocouple locations in the cylindrical castings P3, P6, P8 and P10, and for the TA cups. Chunky graphite relates to solid lines and spiky graphite to dotted lines; note that both were present in casting P10.

**Figure 10 materials-13-05402-f010:**
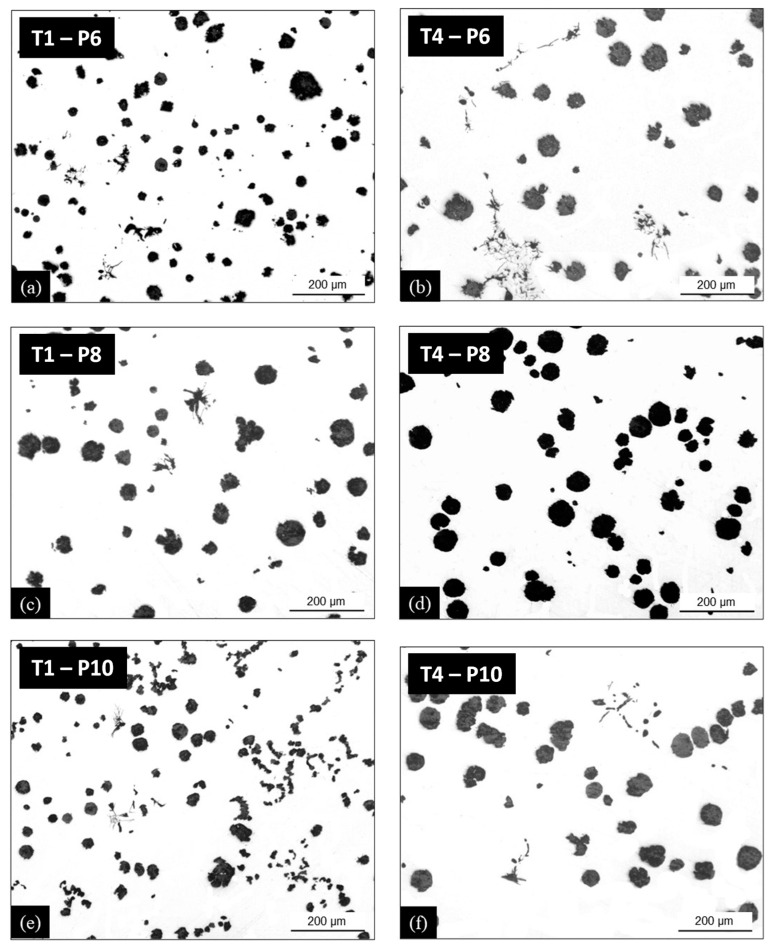
Micrographs of castings P6 (**a**,**b**), P8 (**c**,**d**) and P10 (**e**,**f**), locations T1 (column to the left, (**a**,**c**,**e**)) and T4 (column to the right, (**b**,**d**,**f**)).

**Figure 11 materials-13-05402-f011:**
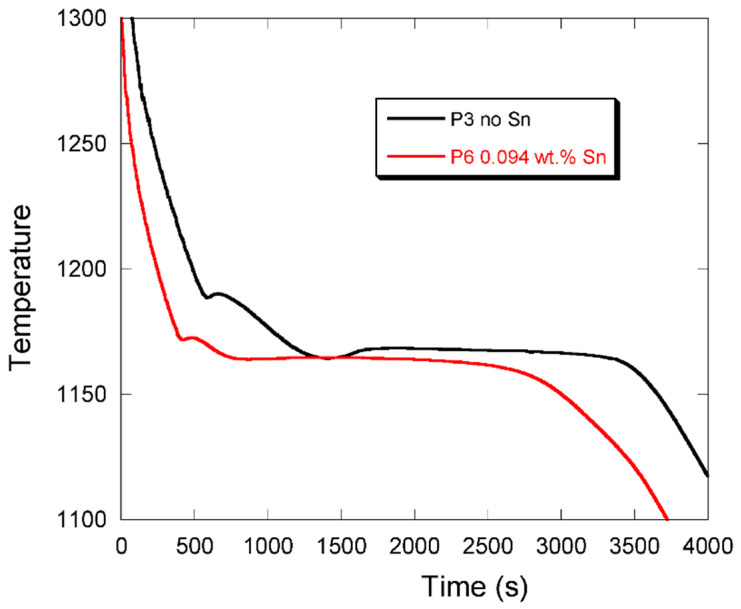
Cooling curves recorded by thermocouple T3 in castings P3 (no Sn added) and P6 (0.094 wt.% Sn).

**Figure 12 materials-13-05402-f012:**
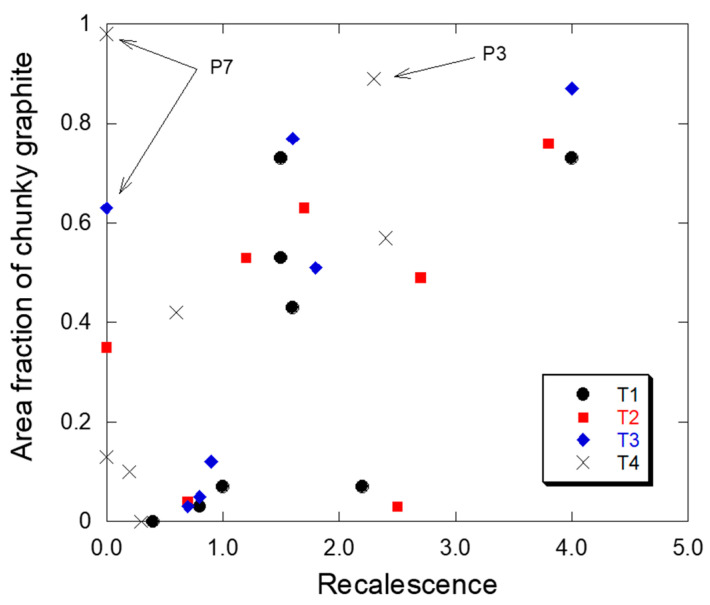
The relation between chunky graphite area fraction and recalescence in the four locations T1–T4 of the cylindrical castings.

**Figure 13 materials-13-05402-f013:**
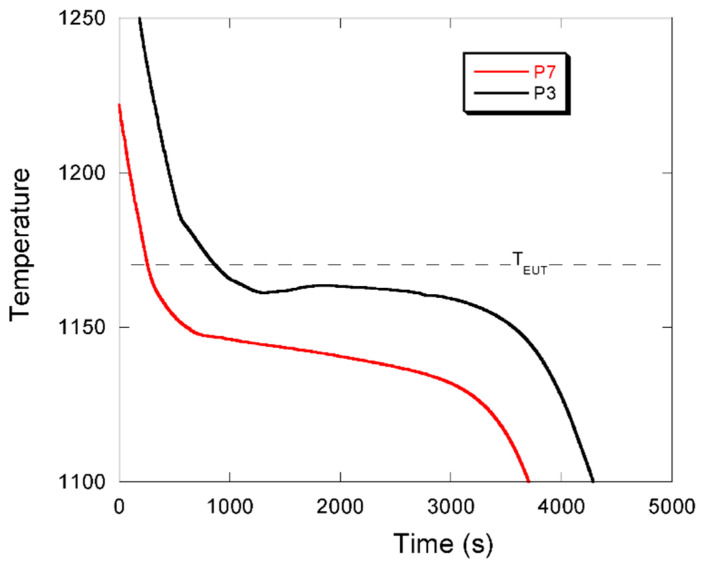
Cooling curves of castings P3 and P7 in location T4.

**Figure 14 materials-13-05402-f014:**
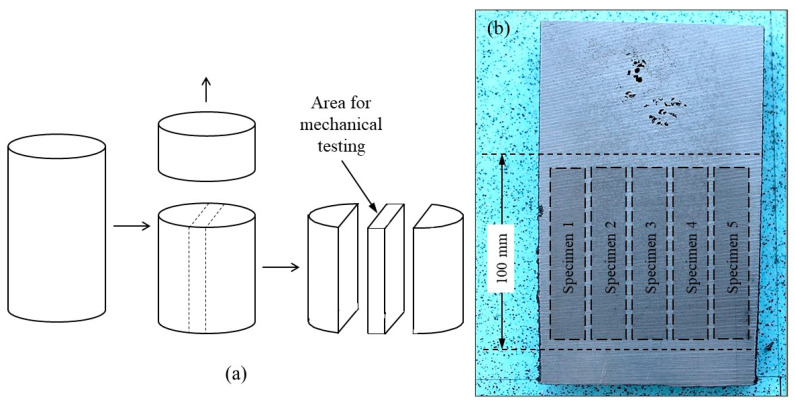
The area used for tensile testing (**a**) and location of the specimens in this area (**b**).

**Figure 15 materials-13-05402-f015:**
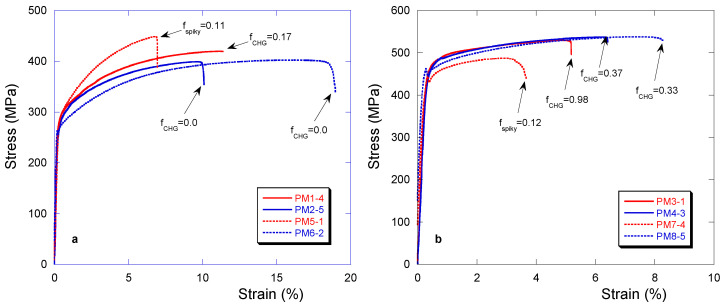
Best stress/strain curves for the low-silicon (**a**) and high-silicon (**b**) castings. The amount of chunky (CHG) or spiky graphite is indicated for each sample along the corresponding curve.

**Figure 16 materials-13-05402-f016:**
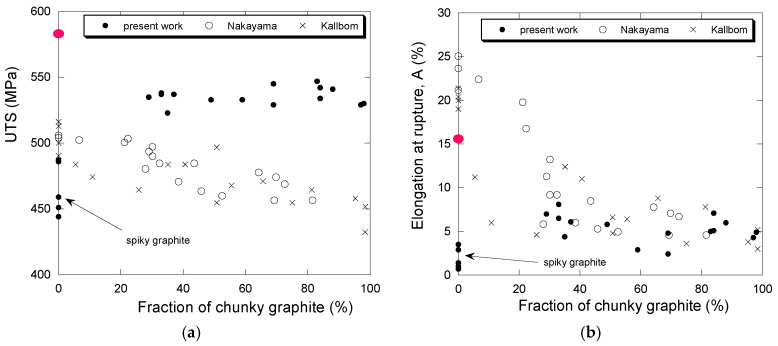
Change of UTS (**a**) and A (**b**) as a function of the fraction of chunky graphite (%) for the high-silicon alloys. Solid dots are for results from the present work, open dots from Nakayama et al. [46] and crosses from Källbom et al. [47]. Outliers in the present results are due to spiky graphite (arrow). The large red dots along the Y-axes locate the value of UTS and A for a 4.2 wt.% Si cast iron without chunky graphite as previously reported [32].

**Figure 17 materials-13-05402-f017:**
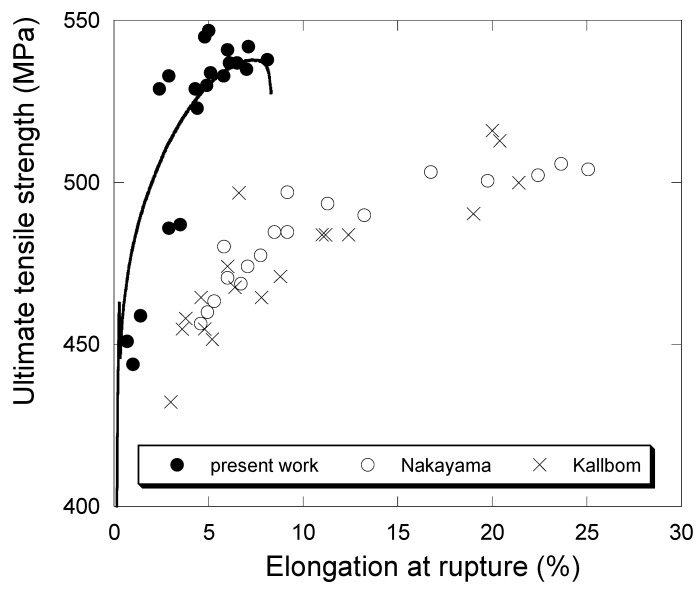
The plot of UTS values versus A values for high silicon alloys from the present work and from literature. The solid curve is the tensile curve for sample PM8-5.

**Table 1 materials-13-05402-t001:** Description of the trials.

Trial	Alloy	FeSiMg	Inoculant	Observation
P2	High Si ductile iron	with RE	Anoc	-
P3		no RE	FeSi75 (no RE)	-
P4	Low Si ductile iron	with RE	Anoc	-
P5		no RE	FeSi75 (no RE)	-
P6	High Si ductile iron	with RE	Anoc	+0.100 wt.% Sn
P7		no RE	FeSi75 (no RE)	Intermediate level of RE
P8				+0.050 wt.% Sn
P9				+0.004 wt.% Sb
P10				+0.025 wt.% Sn

**Table 2 materials-13-05402-t002:** Composition (wt.%) of the FeSi75 alloy for adjustment of Si, of the FeSiMg spheroidizers, and of the Anoc inoculant.

Product	FeSi75	FeSiMg (No RE)	FeSiMg (with RE)	Anoc
Size (mm)	0.2–0.7	5–25	5–25	0.2–0.5
Si	74.40	43.20	46.60	69.90
Mg	-	6.26	6.00	-
Al	0.95	0.25	0.71	0.93
C	0.25	-	-	-
Ca	1.53	0.83	0.96	1.38
Ce	-	<0.05	-	-
La	-	0.16	-	-
RE	-	-	0.92	0.37
Bi	-	-	-	0.49

**Table 3 materials-13-05402-t003:** Classification criteria which are used to determine the different area fractions.

Class	Circularity	Aspect Ratio
III	≤0.70	≥2.0
	<0.50	1.5–2.0
	≤0.35	1.0–1.5
V	>0.70	≥2.0
	≥0.50	1.5–2.0
	0.35–0.77	1.0–1.5
VI	≥0.77	1.0–1.5

**Table 4 materials-13-05402-t004:** Chemical analysis of the castings (wt.%).

Trial	P2	P3	P4	P5	P6	P7	P8	P9	P10
C	3.11	3.01	3.77	3.72	3.04	2.92	3.00	3.03	3.08
Si	3.97	3.91	2.27	2.31	4.20	4.00	4.20	4.10	4.07
Mn	0.17	0.18	0.19	0.18	0.13	0.18	0.21	0.20	0.19
P	0.021	<0.010	<0.010	<0.010	<0.010	0.010	<0.010	<0.010	<0.010
S	0.005	0.006	0.006	0.006	0.006	0.008	0.007	0.008	<0.005
Cr	0.035	0.027	0.036	0.039	0.030	0.036	0.035	0.041	0.0029
Ni	0.051	0.038	0.044	0.041	0.083	0.045	0.046	0.079	0.0038
Cu	0.025	<0.020	0.030	0.052	<0.020	0.026	0.021	0.038	0.0022
Ti	0.018	0.008	0.007	0.007	0.016	0.007	0.008	0.006	0.011
Mg	0.037	0.040	0.046	0.041	0.044	0.039	0.036	0.040	0.045
Ce	0.0067	<0.0005	0.0100	<0.0005	<0.0005	0.0034	<0.0005	<0.0005	<0.0005
La	0.0040	<0.0005	0.0055	<0.0005	<0.0005	0.0017	<0.0005	<0.0005	<0.0005
Sb	<0.0005	<0.0005	<0.0005	<0.0005	<0.0005	<0.0005	<0.0005	0.0039	<0.0005
Sn	<0.005	<0.005	<0.005	<0.005	0.094	<0.005	0.051	<0.005	0.023
Al	0.007	0.008	0.007	0.008	0.009	0.008	0.009	0.009	0.008
CE_99_	4.23	4.11	4.41	4.37	4.22	4.05	4.18	4.18	4.23
CE_ASM_	4.35	4.22	4.47	4.44	4.34	4.16	4.30	4.30	4.34

**Table 5 materials-13-05402-t005:** Castings P2–P5. Average of measured values for total area fraction of graphite (f_A_), relative area fraction for each type of graphite particles (f_III_A_, f_V_A_ and f_VI_A_) and total particle count, N_count_. Type III particles correspond to chunky graphite. Between brackets are listed the minimum and maximum values of f_III_A_ and the standard deviation for N_count_.

Trial	Zone	f_A_	f_III_A_	f_V_A_	f_VI_A_	N_count_
P2	Cup	0.11	0.03 (0.01–0.06)	0.20	0.77	369 (22)
	T1	0.09	0.73 (0.08–1.00)	0.10	0.17	617 (287)
	T2	0.08	0.63 (0.07–1.00)	0.10	0.27	509 (152)
	T3	0.08	0.77 (0.36–1.00)	0.08	0.15	670 (229)
	T4	0.09	0.57 (0.01–1.00)	0.29	0.14	451 (229)
P3	Cup	0.09	0.10 (0.06–0.15)	0.21	0.69	360 (13)
	T1	0.10	0.73 (0.63–0.82)	0.05	0.22	530 (102)
	T2	0.10	0.76 (0.51–0.96)	0.07	0.17	520 (43)
	T3	0.09	0.87 (0.69–1.00)	0.06	0.07	687 (150)
	T4	0.09	0.89 (0.80–0.95)	0.09	0.02	526 (69)
P4	Cup	0.11	0.00	0.21	0.79	354 (29)
	T1	0.11	0.07 (0.00–0.20)	0.55	0.38	133 (31)
	T2	0.10	0.35 (0.07–0.66)	0.40	0.25	170 (35)
	T3	0.11	0.12 (0.02–0.22)	0.56	0.32	128 (24)
	T4	0.11	0.10 (0.00–0.33)	0.52	0.38	152 (55)
P5	Cup	0.10	0.00	0.22	0.78	276 (67)
	T1	0.11	0.43 (0.01–0.98)	0.16	0.41	333 (142)
	T2	0.12	0.49 (0.00–0.87)	0.08	0.43	330 (104)
	T3	0.11	0.51 (0.00–0.95)	0.19	0.30	315 (189)
	T4	0.09	0.42 (0.00–0.88)	0.17	0.41	254 (110)

**Table 6 materials-13-05402-t006:** Castings P7 and P9. Average of measured values for total area fraction of graphite (f_A_), relative area fractions for each type of graphite particles (f_III_A_, f_V_A_ and f_VI_A_) and total particle counts, N_count_. f_III_A_ is reported as either f_CHG_A_ (P7) or f_spiky_A_ (P9). Between brackets are listed the minimum and maximum values of f_III_A_ and the standard deviation for N_count_.

Trial	Zone	f_A_	f_CHG_A_	f_spiky_A_	f_V_A_	f_VI_A_	N_count_
P7	Cup	0.12	0.12 (0.10–0.15)	0.00	0.22	0.66	371 (44)
	T1	0.14	0.53 (0.02–0.87)	0.00	0.16	0.32	384 (155)
	T2	0.11	0.53 (0.32–0.97)	0.00	0.20	0.27	405 (152)
	T3	0.12	0.63 (0.52–0.81)	0.00	0.14	0.23	514 (125)
	T4	0.11	0.98 (0.88–1.00)	0.00	0.01	0.01	742 (136)
P9	Cup	0.12	0.00	0.00	0.23	0.77	380 (39)
	T1	0.13	0.00	0.15 (0.07–0.23)	0.26	0.59	136 (13)
	T2	0.11	0.00	0.08 (0.05–0.13)	0.40	0.52	107 (13)
	T3	0.11	0.00	0.20 (0.08–0.38)	0.39	0.41	100 (12)
	T4	0.11	0.00	0.08 (0.00–0.15)	0.61	0.31	52 (8)

**Table 7 materials-13-05402-t007:** Castings P6, P8 and P10. Average of measured values for total area fraction of graphite (f_A_), relative area fractions for each type of graphite particles (f_III_A_, f_V_A_ and f_VI_A_) and total particle counts, N_count_. f_III_A_ is split in either or both f_CHG_A_ and f_spiky_A_ fractions. Between brackets are listed the minimum and maximum values of f_III_A_ and the standard deviation for N_count_.

Trial	Zone	f_A_	f_CHG_A_	f_spiky_A_	f_V_A_	f_VI_A_	N_count_
P6	Cup	0.08	0.00	0.04 (0.01–0.08)	0.23	0.73	434 (33)
	T1	0.08	0.00	0.03 (0.00–0.06)	0.57	0.40	185 (20)
	T2	0.10	0.00	0.04 (0.01–0.08)	0.75	0.21	162 (20)
	T3	0.06	0.00	0.05 (0.02–0.11)	0.70	0.25	188 (52)
	T4	0.10	0.00	0.13 (0.00–0.30)	0.79	0.08	129 (58)
P8	Cup	0.12	0.00	0.01 (0.00–0.04)	0.15	0.84	349 (9)
	T1	0.10	0.00	0.01 (0.00–0.05)	0.43	0.53	84 (6)
	T2	0.11	0.00	0.01 (0.00–0.01)	0.42	0.56	128 (16)
	T3	0.13	0.00	0.01 (0.00–0.02)	0.37	0.62	119 (16)
	T4	0.12	0.00	0.03 (0.00–0.14)	0.47	0.45	74 (28)
P10	Cup	0.10	0.00	0.00	0.17	0.83	408 (24)
	T1	0.12	0.35 (0.00–0.55)	0.07 (0.00–0.24)	0.14	0.44	210 (44)
	T2	0.12	0.40 (0.05–0.64)	0.03 (0.00–0.06)	0.12	0.45	230 (50)
	T3	0.14	0.78 (0.59–0.96)	0.03 (0.00–0.12)	0.09	0.10	331 (95)
	T4	0.11	0.00	0.01 (0.00–0.04)	0.40	0.59	66 (10)

**Table 8 materials-13-05402-t008:** Composition, tensile testing results and microstructure characteristics of the alloys. f_CHG_A_ stands for area fraction of chunky graphite, f_Spiky_A_ stands for area fraction of spiky graphite, f_A_ stands for total graphite area fraction and N_count_ stands for graphite count. Ferrite and pearlite contents are reported in Appendix A, Table A2, for reference.

Alloy	Composition (wt.%)					Tensile Tests (MPa, A in %)				Microstructure (Average of Three Fields)			
	C	Si	Ce	La	Sb	Sample	UTS	YS	A	f_CHG_A_	f_Spiky_A_	f_A_	N_count_
PM1	3.78	2.46	<0.0005	<0.0005	<0.0005	1	315	285	1.0	0.78	0.00	0.16	614
						2	372	286	5.5	0.00	0.00	0.15	47
						3	335	274	3.0	0.22	0.00	0.12	185
						4	420	283	11.0	0.17	0.00	0.13	166
						5	402	284	6.0	0.00	0.00	0.14	44
PM2	3.88	2.47	0.0041	0.0021	<0.0005	1	367	282	8.0	0.06	0.00	0.12	136
						2	375	296	8.0	0.30	0.00	0.13	165
						3	381	271	10.0	0.10	0.00	0.14	131
						4	400	299	12.5	0.00	0.00	0.14	84
						5	399	280	10.0	0.00	0.00	0.16	91
PM3	3.15	4.17	<0.0005	<0.0005	<0.0005	1	530	470	5.0	0.98	0.00	0.12	705
						2	533	474	3.0	0.59	0.00	0.12	660
						3	529	470	2.5	0.69	0.00	0.12	577
						4	545	466	5.0	0.69	0.00	0.12	574
						5	529	468	4.5	0.97	0.00	0.13	564
PM4	3.20	4.20	0.0038	0.0020	<0.0005	1	547	468	5.0	0.83	0.00	0.15	455
						2	534	466	5.0	0.84	0.00	0.14	493
						3	537	467	6.0	0.37	0.00	0.13	264
						4	541	470	6.0	0.88	0.00	0.13	476
						5	542	462	7.0	0.84	0.00	0.14	379
PM5	3.79	2.44	<0.0005	<0.0005	0.0079	1	449	295	6.5	0.00	0.11	0.14	126
						2	445	296	6.5	0.00	0.06	0.12	104
						3	449	298	5.5	0.00	0.05	0.14	95
						4	425	291	6.5	0.00	0.18	0.14	105
						5	422	291	5.5	0.00	0.06	0.12	97
PM6	3.80	2.43	0.0075	0.0040	0.0072	1	393	268	11.5	0.01	0.00	0.13	121
						2	402	268	19.0	0.00	0.00	0.18	137
						3	396	292	12.5	0.02	0.00	0.15	119
						4	396	270	13.0	0.07	0.00	0.16	101
						5	395	270	13.0	0.16	0.00	0.14	98
PM7	3.27	4.20	<0.0005	<0.0005	0.0043	1	444	441	1.0	0.00	0.24	0.12	128
						2	451	444	0.5	0.00	0.28	0.13	88
						3	459	425	1.5	0.00	0.27	0.15	85
						4	487	430	3.5	0.00	0.12	0.12	144
						5	486	430	3.0	0.00	0.18	0.11	98
PM8	3.22	4.22	0.0073	0.0039	0.0057	1	523	455	4.5	0.35	0.22	0.12	269
						2	537	453	6.5	0.33	0.04	0.13	144
						3	533	480	6.0	0.49	0.08	0.15	230
						4	535	474	7.0	0.29	0.00	0.15	166
						5	538	447	8.0	0.33	0.00	0.12	160

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
