# Peer review of "Chunky Graphite in Low and High Silicon Spheroidal Graphite Cast Irons–Occurrence, Control and Effect on Mechanical Properties"

_materials, 2020, doi:10.3390/ma13235402_

Round 1
Reviewer 1 Report
The article under review can be considered an excellent new piece of research work in the field of ductile cast iron. The subject delt with is of particular interest since there is a significant potential economic impact of the Chunky graphite degeneration in DCI castings, especially in heavy-sectioned components. Presentation of the experiments is of outstanding quality. The reader will benefit from the data listed in detail.
In the following some minor points will be addressed in detail in order to be considered by the authors:
Page 3, line 99: The solidification time of 1 hour is rated to be representative of heavy section castings. Rather 1 hour seems to be a moderate cooling time when solidification times of up to 10 hours for very heavy DCI components with wall thicknesses of up to 500 mm are kept in mind. Please consider changing the formulation.
Page 5, line 146: The area fractions of spiky and chunky graphite were separately determined when both types were present at the micrograph. Please comment on how this was practically done since the same classification criteria for class III (chunky and spiky) have been applied (Table 3).
Page 7, lines 197-199: This is an important point which illustrates the inherent difficulties in quantifying the Chunky graphite amount due to its 3d irregular non-isotropic and non-dispers structure. Example: In P2 T4 the scatter in f_III_A results ranges from 1 % to 100 % of CHG. Please comment on the implications of these features on the interpretation of the experimental results.
Page 8, line 230: Please be careful in rating the decrease in chunky graphite significant with the high Si alloys. With T1 there is a difference of 0%, with T2 and T3 its about 10 %. Keeping the scatter in local measurements in mind, this can be rated more a trend but not significant.
Page 9, Fig. 6: Please check the unit at the Y-axis.
Page 9, line 248: The difference between P2 and P3 is only 10% of CHG. Again, please keep the big scatter in local measurements in mind.
Page 10, line 272: Please check to place "exists" after "RE".
Page 14, Fig. 11: Please check the unit at the Y-axis.
Page 15, Fig. 12: Please check the unit at the X-axis.
Page 16, Fig. 14: Is there an influence of the radial specimen position due to differences in the cooling and solidification rate across the diameter of the cylinder? Has this been considered with respect to the resulting mechanical properties?
Page 16, line 407: Please describe where the microstructural characteristics have been determined in or on the specimens. This information is missing but of high importance for the significance of the microstructural results with respect to the mechanical properties.
Page 17, Table 7: Please check the layout of the YS column. Should be a little broader to avoid the wraparound.
Page 17, Table 7: Column A: The measured values must be given rounded to the nearest 0.5%. This is the regulation of the relevant testing standard ISO 6892-1 for the parameter elongation after rupture. The background of this is the usual measurement uncertainty with A.
Page 18, lines 422-423: The assessment is based on one specimen only of each condition. Please consider to mention this at this point.
Page 18, line 426: Sentence could be extended at the end to "... higher tensil strength and decreasing elongation of rupture.".
Page 19, line 440: The rating "certainly overestimated" may be seen as a little misleading. Values measured at fracture surfaces may simply be different from those determined at microsections anywhere in the specimen or component. Mostly they will be higher at the fracture surface. But this simply points at the heterogeneous nature of CHG and the failure mechanism according to which the fracture follows the path of least energy. The question of a adequate measure of CHG contents and especially its determination is unfortunately still open.
Page 20, lines 465-467: It would be informative for the reader if the sample sizes were given.
Page 20, line 473: Nakayima --> Nakayama
Page 21, line 508: Please check if Foglio did really concluded this for quasi-static tensile properties. The focus of his work cited here was on fatigue investigations.
Page 22, line 540: Please consider to modify your judgement "much more" keeping the reviewers comments on scatter of CHG and spiky graphite vs. number and location of examination fields in mind.
Reviewer 2 Report
1. What is the dimension of the molds that the authors used? How was the molds isolated? The amount of spheroidizer and inoculant added was not given.
2. K-type thermocouples with a metallic sheath 1.5 mm in diameter were employed in P8 trial while keeping the same experimental method. Why?
3. How is the total particle count (Ncount) clculated? Especially for the Chunky graphite . And the standard deviation for Ncount?
4. When using the sheathed K-type thermocouples, there is a lag in for the collection of temperatures. It is means that the temperatures collected is far below the the real temperatures, especially when the cooling rate is big, for example T4 in this manuscript. I want to know how do the authors consider about this? And the influences of the inaccurate temperatures on the analysis based on that.
Reviewer 3 Report
In my opinion the aricle is very interesting, for printing after minor corrections:
Line 666, The figure A1. Unreadsbly described picture micrograph . Unify the signatures.
Reviewer 4 Report
A well-planned, well-executed and well-described experiment. The authors performed a very systematic analysis of the influence of rare earth and silicon impurities on the emission of graphite chunky in cast iron. The strength of the work is also the analysis of the interaction of these impurities with components of Sn and Sb alloys.
Proper selection of methods, microstructures (with detailed quantitative metallographic analysis), cooling curves, EDX maps and tensile curves are presented.
Consider reducing the font in Table 7 or giving the Ce + La concentration together.
The units on the coordinate axes in Figures 6, 11, 12, 13 should be corrected.
